# Nanofiltration Composite Membranes Based on KIT-6 and Functionalized KIT-6 Nanoparticles in a Polymeric Matrix with Enhanced Performances

**DOI:** 10.3390/membranes11050300

**Published:** 2021-04-21

**Authors:** Gabriela Paun, Viorica Parvulescu, Elena Neagu, Camelia Albu, Larisa Ionita, Monica Elisabeta Maxim, Andrei Munteanu, Madalina Ciobanu, Gabriel Lucian Radu

**Affiliations:** 1National Institute for Research-Development of Biological Sciences, 060031 Bucharest, Romania; gpaunroman@gmail.com (G.P.); lucineagu2006@yahoo.com (E.N.); camelia_barsan2000@yahoo.com (C.A.); larisa.calu@yahoo.com (L.I.); 2Ilie Murgulescu Institute of Physical Chemistry of the Romanian Academy, Spl. Independentei 202, 060021 Bucharest, Romania; vpirvulescu@icf.ro (V.P.); monimaxim@gmail.com (M.E.M.); madalinabesnea@yahoo.com (M.C.); 3Apel Laser, 25 Vanatorilor Street, 077135 Mogosoaia, Romania; andrei.munteanu@apellaser.ro

**Keywords:** nanofiltration composite membranes, PPEES, KIT-6, KIT-6 functionalized, selective separation

## Abstract

The nanofiltration composite membranes were obtained by incorporation of KIT-6 ordered mesoporous silica, before and after its functionalization with amine groups, into polyphenylene-ether-ether-sulfone (PPEES) matrix. The incorporation of silica nanoparticles into PPEES polymer matrix was evidenced by FTIR and UV–VIS spectroscopy. SEM images of the membranes cross-section and their surface topology, evidenced by AFM, showed a low effect of KIT-6 silica nanoparticles loading and functionalization. The performances of the obtained membranes were appraised in permeation of *Chaenomeles japonica* fruit extracts and the selective separation of phenolic acids and flavonoids. The obtained results proved that the PPEES with functionalized KIT-6 nanofiltration membrane, we have prepared, is suitable for the polyphenolic compound’s concentration from the natural extracts.

## 1. Introduction

Particular interest has been given in recent years to finding solutions to improve the characteristics of membranes, such as a high permeate flux, rejection, and better antifouling capacity. The most polymeric membranes are hydrophobic nature, which causes serious fouling problems, leading to in a permanent flux decline, shortened membrane lifetime, and increasing the maintenance costs [1]. Previous studies indicated that the physical–chemical properties of membrane surface, such as hydrophilicity and roughness, are major factors influencing membrane fouling. It is generally accepted that hydrophilic membrane corresponds to lower membrane fouling potential than hydrophobic one because many foulants are hydrophobic compounds [2,3]. In addition, the most hydrophobic nanofiltration membrane provided the lowest permeate flux in the nanofiltration extract [4].

In order to reduce these limitations and to obtain high performance polymeric membranes, [5] dispersed inorganic nanoparticles into a polymer matrix, forming so called polymeric nanocomposite membranes [6,7,8,9]. The obtained membranes show superior performances, such as mechanical toughness, thermal properties, permeability, and selectivity, compared to polymeric membranes. All of these depend on the quality of interface between nanoparticles and the polymer. Thus, better performing membranes containing smaller size inorganic particles (<20 nm) were obtained [10]. Moreover, the functionalization of these nanoparticles increases their dispersion and the hydrophilic characteristics of the membrane. As well, the previous studies demonstrated that inorganic nanoparticles such as silica (SiO_2_) [11,12,13,14], zeolites [15,16], metal oxide (Al_2_O_3_, TiO_2_, ZnO) [15,17,18,19,20,21,22,23], and carbon nanotubes [23,24] could improve the hydrophilicity of the modified polymer hydrophobic membranes. Silica nanoparticles have chemical compatible and the mechanical stability needed for preparation of polymeric membrane, and their structure can be changed by chemical functionalization [25]. Thus, SBA-15 ordered mesoporous silica was used as nanofiller to improve the water permeability of a polymer matrix [26]. A higher compatibility and adhesion between inorganic nanoparticles and polymer matrix were obtained by functionalization with amine groups of silica surface in the presence of (3-aminopropyl) triethoxysilane (APTES) as coupling agent.

Another method to improve hydrophilicity of the polymer membranes is the addition of polyvinyl pyrrolidone (PVP) or polyethylene glycol (PEG) in different concentrations [27,28]. 

Polyphenylene ether ether sulfone (PPEES) was studied little until 2015 for the membrane synthesis. Our previous studies showed that PPEES can be successfully applied to obtain ultrafiltration membranes for concentration of the herbal extract [29]. PPEES received in the last time more attention due to the specific properties, such as less hydrophobicity, good solubility, good electronic properties, high thermal resistance and a good chemical stability [30,31]. An interesting application was as cross-linkers, to induce the polymer network with interpenetrating structure, and at the same time, as a spreader to homogeneously disperse SiO_2_ nanoparticles in nanocomposite membranes [32]. The chemical structure of PPEES is similar to the polysulfone’s structure, so that the properties of these two polymers are similar, but PPEES is much cheaper and therefore, it is preferable to replace polysulfone with poly(1,4-phenylene-ether-sulfone-ether).

Here, we report on incorporation of KIT-6 ordered mesoporous silica, before and after its functionalization with amine groups, into polyphenylene-ether-ether-sulfone matrix. KIT-6 was selected as nanofiller due to the properties such as high surface area, ordered porous structure, mesopores with narrow pore sizes, high adsorption capacity, and thermal and hydrothermal stability [33,34]. The properties of membranes and the influence of incorporation and composition of KIT-6 and KIT-6-NH_2_ nanoparticles on permeation and selectivity of phenolic acids and flavonoids of *Chaenomeles japonica *fruit extracts were studied.

## 2. Materials and Methods

### 2.1. Synthesis of KIT-6 and Functionalized KIT-6-NH_2_ Mesoporous Silica Nanoparticles

KIT-6 silica nanoparticles were prepared using Pluronic 123, as surfactant, by dispersing in water, hydrochloric acid, and butanol [35]. The silica precursor, i.e., tetraethyl orthosilicate (TEOS) was added under stirring, and the resulted gel was transferred into autoclave to treat hydrothermally for 48 h at 80, 100, or 120 °C. The obtained solids were filtered, washed, and treated thermally at 100 °C, for drying, and at 550 °C to remove the surfactant.

The functionalization with amine group of the KIT-6 mesoporous silica surface was accomplished by post-grafting method [36,37] with 3-aminopropyltriethoxysilane (APTES) as coupling agent.

### 2.2. Preparation of Nanofiltration Membranes

The nanofiltration composite membranes were obtained by phase inversion method. Poly(1,4-phenylene ether-ether-sulfone) (PPEES; Sigma-Aldrich Chemical Company, Inc., St. Louis, USA) was dissolved in 1-Methyl-2-pyrrolidone (NMP; Honeywell, SUA). The silica nanoparticles and polyvinyl pyrrolidone K90 were added in the polymeric solution under continuous stirring to form a casting solution. The resulting homogeneous solution was subjected to sonication for 45 min to remove trapped bubbles, and then, it was placed as thin layer with a “doctor blade” roller. The polymer was precipitated in a bath with deionized water. The obtained nanocomposite membranes were kept for 3–4 h in deionized water to afford complete phase separation. The membranes with various composition of inorganic nanoparticles were named “Mn” (n = 0, 1–4). The composition of different casting solutions is presented in Table 1.

### 2.3. Characterization of the Obtained Materials 

The structural and textural properties of KIT-6 and KIT-6-NH_2_ nanoparticles were evaluated by X-ray diffraction at small angles (Bruker AXS D8 diffractometer, Mannheim, Germany) and by N_2_ adsorption–desorption (Micromeritics ASAP 2010, Dresden, Germany). The morphology and ordered porous structure were characterized by scanning electron microscopy (SEM with EDX, FEI Quanta 3D FEG), and transmission electron microscopy (TEM, Tecnai 10 G2-F30) from FEI Company Europe, Eindhoven, Netherlands).

The morphology of the polymeric and nanocomposite membranes cross-section was analyzed by scanning electron microscopy (SEM-Hitachi model SU1510, Schaumburg, IL, USA). The SEM operating conditions were 15 kV, with a nominal probe current reading of 30 nA, and a working distance of 15 mm. The samples were left uncoated (all images were taken in variable pressure mode at 30 Pa to compensate the surface chargeup effect so that coating with a conductive layer was not necessary) and the SEM was operated in SEM high-vacuum mode (chamber pressure 30 Pa), using an atmosphere of dry nitrogen gas. 

The surface roughness of the prepared membranes was analyzed by atomic force microscopy (AFM CORE Nanosurf, Liestal, Switzerland). The chemistry of membranes surface was studied by FTIR spectroscopy with Bruker TENSOR 27 instrument (Mannheim, Germany). FTIR spectra were recorded between 400 and 4000 cm^−1^. The UV–VIS diffuse reflectance spectra were recorded on a JASCO V570 spectrophotometer (ABL&E Jasco Romania SRL, Cluj-Napoca, Romania). As a reference, a certified reflectance standard, spectral, was used, and the measurements were carried out in the range of 60–190 nm.

The water contact angle measurement using a Drop Shape Analysis System, model DSA 2 Easy Drop instrument (Krüss GmbH, Germany) is an evaluation of the membrane’s hydrophilicity. The contact angle (*θ*) values of each sample were measured at five diverse positions of one sample.

### 2.4. Permeability and Selectivity of Membranes

Performances of the prepared membranes were assessed by a KMS Laboratory Cell CF-1 cross-flow lab-scale filtration unit. The obtained results for nanocomposite membranes were compared with that of the unfilled one.

The permeability and selectivity were evaluated using *Chaenomeles japonica *fruits extract. The permeate flux (*J*) and rejection rate (*R_j_*) was calculated with equations:(1)$$J=\frac{V}{A\cdot t}\;(L\;m^{-2}\;h^{-1}),$$
where *V* is the permeate volume (L), *A* is the effective membrane area (m^2^), and *t* is the time (h) necessary for the V litters of permeate to be collected.

The experiment ended when a volumetric concentration ratio (feed volume/retentate volume) of 2.5 was reached. Samples of feed solution, permeate, and retentate were collected for further analysis.

The rejection to solute (*R*, %) was determined through the formula: (2)$$R_{j}=\left(1-\frac{Cp}{Cf}\right)\times 100\;(\%),$$
where *c_p_* and *c_f_* represents concentration in permeate and feed solution, respectively.

The *C. japonica *extract was achieved by accelerate solvent extraction using a Dionex ASE 350 System. The ASE conditions were set as: solvent—60% ethanol, temperature—100 °C, static time—10 min, and static cycle—3.

The analysis of the extract concentration was performed by determination the phenolic acids and flavonoids using spectrophotometric methods and HPLC/MS method. The phenolic content was seated with Folin–Ciocalteu assay [38] and expressed as chlorogenic acid equivalents (CAE) mg/mL).

The total flavonoid content was measured using the AlCl_3_ colorimetric method as described by Lin [39] and was expressed as rutin equivalents (RE) μg/mL.

The HPLC-MS polyphenol measurements were performed by HPLC SHIMADZU system, through a C18 Kromasil 3.5, 2.1 mm × 100 mm column, using a validated HPLC-MS method [40].

## 3. Results and Discussion

### 3.1. Characterizations of the Inorganic Nanoparticles 

The ordered mesoporous structure of KIT-6 (obtained by treatment in autoclave at 100 °C) and KIT-6-NH_2_ nanoparticles, with Ia3d symmetry, was evidenced by small angle XRD patterns (Figure 1a). Although the ordered porous structure was preserved, a small decrease in XRD peaks intensity, pore volume, and pore size can be observed for KIT6-NH_2_. 

The textural properties were characterized by N_2_ sorption experiments. As, we can see that adsorption branch of isotherms (Figure 1b) shows a sharp inflection between 0.6 and 0.8 range of the relative pressure which is typical for mesoporous silica materials such as KIT-6 mesoporous silica [33]. It can also pollable to observe the preservation of KIT-6 texture with H1 hysteresis loop after functionalization of silica with amine group. The samples show a narrow pore size distribution centered at 7.2 and 6.4 nm. 

The adsorption branch of isotherms (Figure 1b) shows a sharp inflection between 0.6 and 0.8 range of the relative pressure, thus highlighting the preservation of KIT-6 texture with H1 hysteresis loop. The samples show a narrow pore size distribution centered at 7.2 and 6.4 nm. The best surface area (780 m^2^/g) was obtained for sample obtained by hydrothermal treatment at 100 °C. The decrease in of volume mesopores relative to KIT-6 silica and the decrease in the pore diameter confirm the location of the grafted species inside the mesopores, and not just on the external surface.

Another property of the inorganic nanoparticles that determines their adhesion with polymer matrix, structure, and stability of the obtained membrane is morphology. Thus, morphology was the main property selected for the obtained silica nanoparticles at different hydrothermal treatment temperature. Thus, SEM images (Figure 2) show larger and more compacted particles for samples obtained by hydrothermal treatment at 80 °C. For higher temperature (120 °C), a larger particle size distribution can be observed. Therefore, the optimum temperature of hydrothermal treatment is 100 °C, condition in which a spherical morphology of particles with smaller and more uniform sizes (Figure 3a) was obtained for KIT-6 powder. 

A high percent of smaller spherical particles was obtained at higher temperature. Figure 3 evidence the morphology (Figure 3a) and the ordered porous structure (Figure 3b) of the selected KIT-6 samples for nanocomposite membrane preparation. After functionalization, the morphology remains unchanged. TEM images show the channels with uniform pore size typically for this type of mesoporous silica.

### 3.2. Characterizations of the Nanofiltration Composite Membranes

SEM micrographs of cross-section of the prepared membranes (Figure 4) exhibit an asymmetric structure, with a finger-like morphology. These results for nanofiltration composite membranes evidenced a uniform surface, denser top layer, and porous sub-layer with macropores for permeation. The SEM images showed that the addition of 1–2% of KIT-6 and KIT-6-NH_2_ nanoparticles in membranes exhibit an asymmetric structure similar to the base PPEES membrane [41]. Adding KIT silica nanoparticles to PPEES polymeric solution led to slight change in the morphology of channels. Decreasing of number and pore sizes can be observed for samples with higher concentration (2%) of silica nanoparticle. The KIT-6 and KIT-6-NH_2_ nanoparticles are found close by and outside the pores forming a low-level roughness on composite membrane surface. 

The results of the surface topography of the membranes, obtained by AFM, are presented in Figure 5. These images show, in condition of low silica concentration, insignificant variation of roughness and surface porosity compared with PPEES membrane (M0). The decrease in membrane roughness surface was observed for nanofiltration composite membranes with amine-functionalized KIT-6 nanoparticles (M2, M4a and M4b, where M4a is the cross-sectional image and M4b is the membrane surface image). This was result of better interaction between inorganic nanoparticles and PPEES polymeric matrix.

The effect of silica nanoparticles on hydrophilicity of microporous PPEES membranes was evaluated by contact angle measurement. Figure 6 presents the water contact angle profiles for the prepared membranes. A low water contact angle indicates a better surface hydrophilicity and water wettability. The results revelated that all nanofiltration composite membranes had a lower contact angle than PPEES membrane, and also, the hydrophilicity of nanofiltration composite membranes based on KIT-6 and functionalized KIT-6 nanoparticles was improved, which suggests enhanced flux and antifouling properties of membranes. Furthermore, a decrease in the contact angle was obtained with increasing the concentration of silica nanoparticles. Our result is in agreement with other similar studies, which showed that the addition of SiO_2_ nanoparticles in the polymeric matrix tended to decrease membrane contact angles [42].

Dispersion of the silica into PPEES polymeric matrix was evidenced by FTIR spectra of the prepared membranes (Figure 7). 

The FTIR spectrum of PPEES and PPEES with silica nanoparticles membrane displayed following results: (a) peaks at 810 cm^−1^ was assigned to Si-O-Si vibration and 960 cm^−1^ was assigned to Si-O stretching vibration of SiOH group corresponding to KIT-6, and KIT-6-NH_2_ appears only on the M1 and M2 membranes; (b) 1107–1072 cm^−1^ was attributed to the aromatic ring vibrations; (c) 1150 cm^−1^ corresponded to the symmetric O=S=O stretching of sulfone group; (d) 1230 cm^−1^ was attributed to the aromatic ether and 1323 cm^−1^ was attributed to the S=O stretching in sulfone; (e) 1412 cm^−1^ corresponded to asymmetric C-H bending deformation of methyl group; (f) 1525 cm^−1^ (N-H bending) indicated the presence of -NH_2_ groups in amino silica; (g) 1590 cm^−1^ was attributed to the C=C aromatic ring vibrations; (h) the band at 1670 cm^−1^ correlated with C=O stretching, which might indicate the existence of PVP in the matrix; and (i) 3600–3400 cm^–1^ corresponded to O-H stretching vibrations.

The presence of inorganic nanoparticles in PPEES matrix and their molecular interactions were evaluated by UV–VIS absorption of composite membranes (Figure 8).

The UV–VIS spectra show that PPEES polymer and polymer with silica nanoparticles, with different concentrations, exhibited similar absorption bands at 258 and 298 nm, indicating that the center of absorption bands is related to intramolecular and intermolecular charge-transfer interactions. The intensities of the absorption bands decrease after the introduction of silica nanoparticles, this being a proof of the interaction between silica and polymer. Furthermore, the strongest interaction can be seen in the case of PPEES polymer and functionalized silica, the strength of the interaction increasing with the amount of functionalized silica (2 wt. %). 

In order to study the efficiency of composite nanofiltration membranes compared to the PPEES membranes, the water flux, permeate flux, and polyphenols and flavonoid rejections were measured (Table 2).

With increasing KIT-6 or KIT-6-NH_2_ silica nanoparticles content from 1 to 2 wt. % and introducing amino group, the separation efficiency for total polyphenols and flavonoids was enhanced sharply. As is presented in Table 2, the PPEES (M0) and composite nanofiltration membranes (M1–M4) present a higher rejection coefficient to total polyphenols compare with flavonoids. The results from Table 2 indicate that membranes with KIT-6-NH_2_ silica nanoparticles showed highest rejection values for polyphenol, with 33% higher than rejection for M0 and also with 103% higher than rejection of flavonoid compounds for M0 membrane. 

These types of composite nanofiltration membranes retain the analyzed compounds more efficiently due to better interaction, i.e., hydrogen bonding between amine group of silica nanoparticles and hydroxyl group of polyphenol and flavonoid compounds. The incorporation of KIT-6 and KIT-6-NH_2_ hydrophilic silica nanoparticles in the polymeric matrix improves the hydrophilicity of the prepared composite membranes, and the flux values increased directly with the quantity of silica nanoparticles from the membrane, which is in agreement with previously published results [37].

Figure 9 displays the linear variation of pure water permeation fluxes with the trans-membrane pressure, for the all-prepared membranes.

The separation performance of the optimal membrane towards different polyphenolic compounds was further studied. The HPLC data for phenolic acids and flavonoids in the ASE extract and retentate using PPEES (M0) and PPEES with silica nanoparticles (M3 and M4) membranes are presented in Table 3. The quantification of these compounds is important because the most phenolic acids and flavones have medicinal importance in treating chronic diseases [38,39,40]. Recent studies proved the efficiency of nanofiltration for concentration of polyphenols from natural extracts [43,44,45].

The results obtained by HPLC confirm the results presented in Table 2 regarding the efficiency and selectivity in the concentration of phenolic acids and flavonoids with KIT-6-NH_2_ composite membranes. 

## 4. Conclusions

Four composite nanofiltration, membranes were obtained by incorporation of KIT-6 ordered mesoporous silica, before and after its functionalization with amine groups, into polyphenylene-ether-ether-sulfone (PPEES) matrix. A low effect of loading and functionalization of KIT-6 silica nanoparticles on the cross-section of membranes and their surface topology was evidenced. The evaluation of *Chaenomeles japonica *fruit extracts’ permeation and selectivity of phenolic acids and flavonoids shows a 33% higher rejection of phenolic acids and also 103% higher rejection of flavonoid compounds for composite membrane with functionalized KIT-6 and highest loading compared to PPEES membrane.

## Figures and Tables

**Figure 1 membranes-11-00300-f001:**
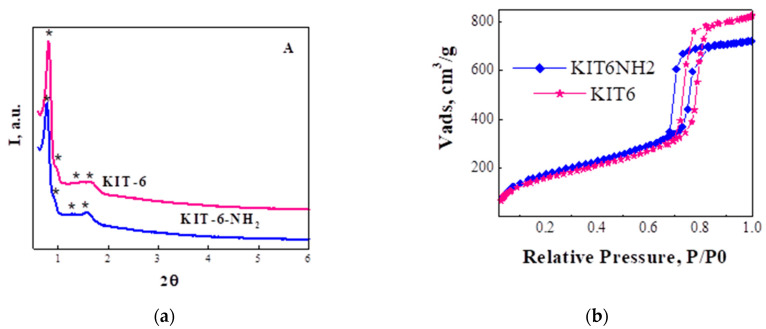
(**a**) XRD patterns at small angle and (**b**) N_2_ adsorption desorption of KIT-6 mesoporous nanoparticles before and after functionalization.

**Figure 2 membranes-11-00300-f002:**
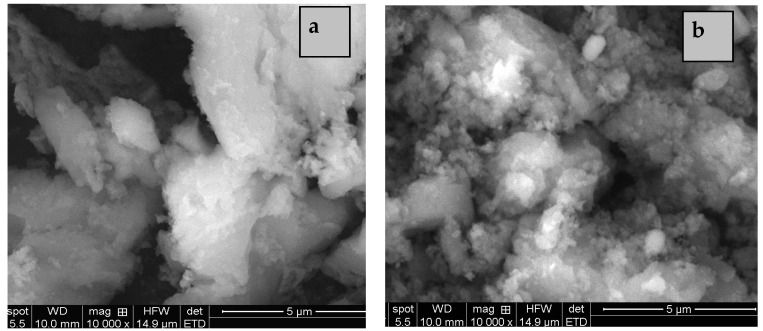
SEM images of KIT-6 mesoporous silica hydrothermal treated at 80 °C (**a**) and 120 °C (**b**).

**Figure 3 membranes-11-00300-f003:**
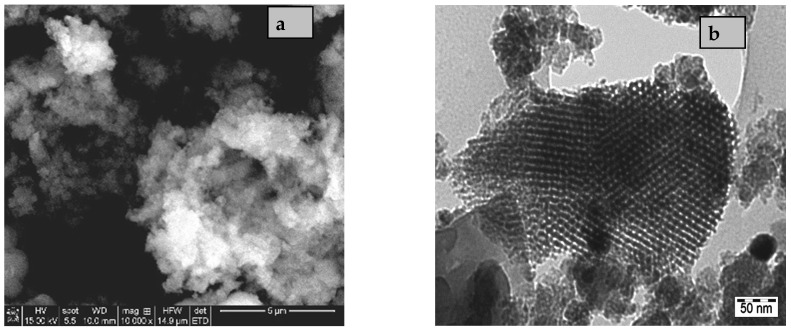
SEM images (**a**) and TEM images (**b**) of the selected KIT-6 sample.

**Figure 4 membranes-11-00300-f004:**
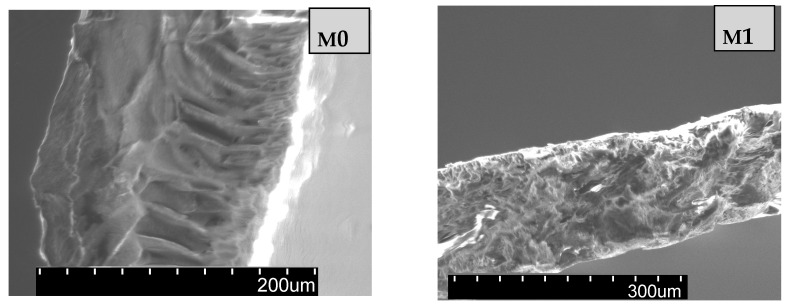
SEM cross-sectional images for the obtained membranes with different concentration of KIT-6 and KIT-6-NH_2_ nanoparticles.

**Figure 5 membranes-11-00300-f005:**
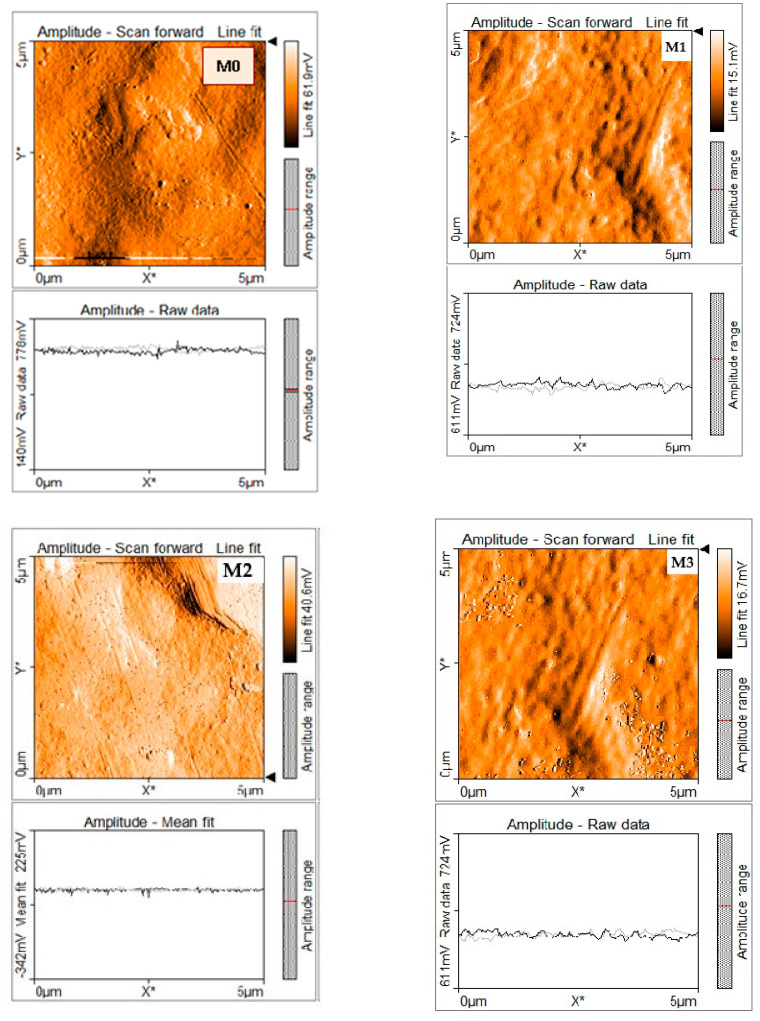
Atomic force microscopy (AFM) images of the obtained membranes with different concentrations of KIT-6 and KIT-6-NH_2_ nanoparticles.

**Figure 6 membranes-11-00300-f006:**
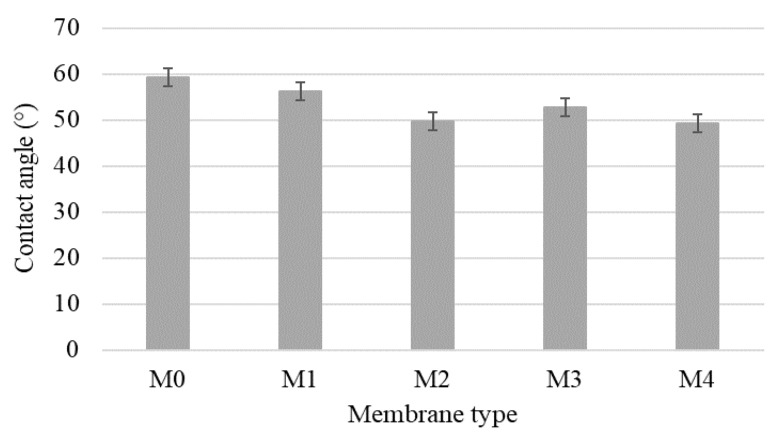
The contact angle of polyphenylene-ether-ether-sulfone (PPEES) and nanocomposite membranes.

**Figure 7 membranes-11-00300-f007:**
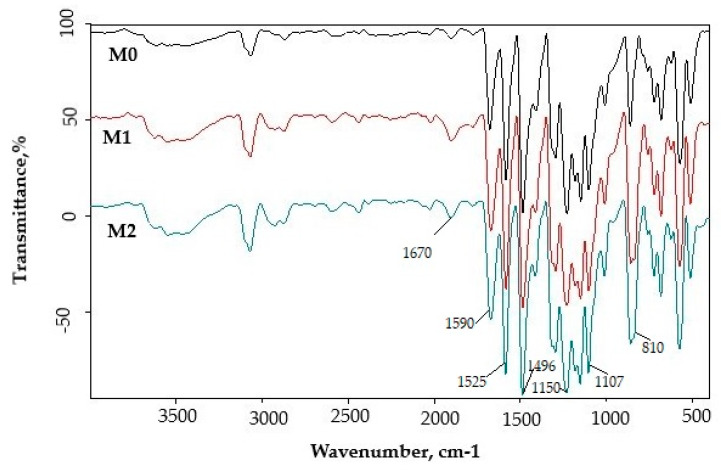
FTIR spectra of PPEES (M0) and PPEES with silica nanoparticles (M1 and M2).

**Figure 8 membranes-11-00300-f008:**
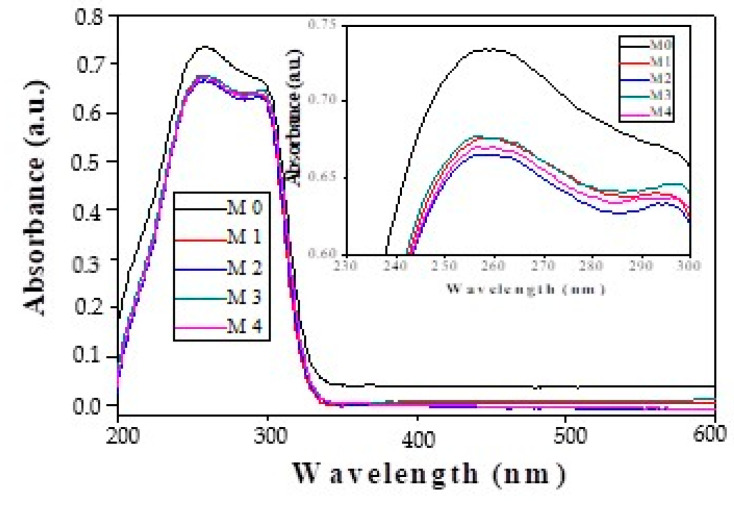
UV–VIS absorption spectrum of PPEES and nanocomposite membranes.

**Figure 9 membranes-11-00300-f009:**
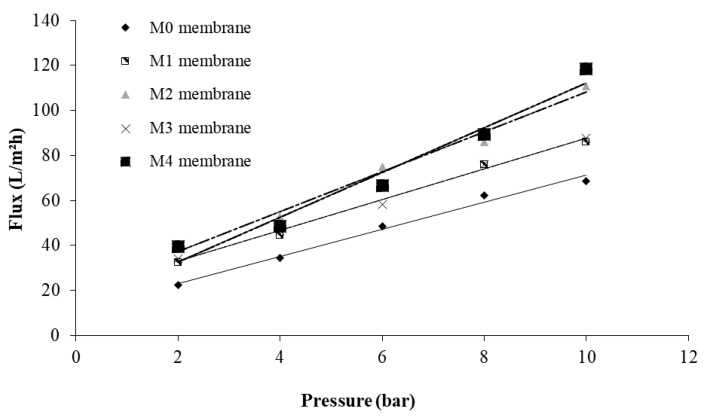
Water flux versus applied pressure for membrane (M0) and composite membranes (M1–M4).

**Table 1 membranes-11-00300-t001:** Compositions of the casting solution.

Membrane Type	Polymer (wt. %)		Silica Nanoparticle (wt. %)	
	PPEES	PVP	KIT-6	KIT-6-NH_2_
M0	20	2	0	0
M1	20	2	1	0
M2	20	2	0	1
M3	20	2	2	0
M4	20	2	0	2

**Table 2 membranes-11-00300-t002:** Permeation performance and rejection for total polyphenols and flavonoids from *Chaenomeles japonica *fruits extract for the prepared membranes.

Membrane Type	Pure Water Flux ^a^ (Lm^−2^h^−1^)	Extract Flux ^a^ (Lm^−2^h^−1^)	Total Polyphenols Rejection (%)	Flavonoid’s Rejection (%)
M0	62.3 ± 0.4	4.1 ± 0.04	60.8 ± 0.5	31.4 ± 0.09
M1	75.9 ± 0.5	6.3 ± 0.05	64.8 ± 0.4	55.0 ± 0.3
M2	86.1 ± 0.7	9.1 ± 0.09	79.5 ± 0.6	60.4 ± 0.5
M3	76.1 ± 0.6	7.7 ± 0.06	79.9 ± 0.6	61.8 ± 0.4
M4	87.5 ± 0.6	10.8 ± 0.09	80.9 ± 0.7	63.8 ± 0.5

^a^ Obtained through filtration of pure water and extract, respectively, at 25 ± 1 °C and 8 bar.

**Table 3 membranes-11-00300-t003:** HPLC-MS analysis results for ASE extract and retentate fractions.

Compound [M/z]^-^	ASE Extract	Retentate (M0)	Retentate (M3)	Retentate (M4)
	μg/mL	μg/mL	μg/mL	μg/mL
**Ellagic acid [301]**	1.41	1.72	1.74	2.30
**Rutin [609]**	1.05	1.17	1.35	1.63
**Quercetin-3-β-D-qlucoside** **(isoquercitrin) [463]**	3.26	4.11	4.12	5.31
**Epicatechin [289]**	9.24	8.61	7.85	9.17
**Quercetol [301]**	0.43	0.42	0.47	0.52
**Myricetin [317]**	0.68	0.69	0.70	0.77
**Chlorogenic acid [353]**	46.62	56.68	69.29	70.09
**Luteolin [285]**	0.25	0.26	0.26	0.29
**4-Hydroxybenzoic acid [137]**	18.03	22.11	24.66	30.66
**Sinapic acid [223]**	1.09	1.06	1.08	1.12

## Data Availability

The data presented in this study are openly available at [**doi**], reference number.

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
