# Peer review of "Nanofiltration Composite Membranes Based on KIT-6 and Functionalized KIT-6 Nanoparticles in a Polymeric Matrix with Enhanced Performances"

_membranes, 2021, doi:10.3390/membranes11050300_

Round 1

Reviewer 1 Report

Please find the attached review comment file. 

Author Response

Response to Reviewer 1 Comments

Point 1. In the title, the first alphabet of word should be a capital.

Response 1: We modified the format of Title.

Point 2. The space needs between number and unit, except the unit of % and ï‚°.

Response 2: The article has been reviewed and all the corrections have been done.

Point 3. On page 3, the font style of equation (1) and (2) is different, compared to the font of main text.

Response 3: The article has been reviewed and all the corrections have been done.

Point 4. In Figure 1, the XRD pattern image is unclear, upgrade the image resolution.

Response 4: We corrected the XRD pattern image (Figure 1) for a better resolution.

Point 5. In Figure 2 ~ Figure 3, the image processing is so messy. For example, the image sizes are individual and the gap between images is different.

Response 5: We replaced figures 2 and 3 so that the dimensions of the images are the same.

Point 6. In Figure 5, AFM images were obtained with the different scan area. AFM images should be remeasured with the same scan area in order to fairly compare each other. Suggest the roughness and surface porosity values of the samples to discuss their surface properties. In addition, the image processing is so messy.

Response 6: We changed the AFM images as the reviewer suggested. 

Point 7. Send the Figure 7 to supporting data or merge the results of Figure 6 and 7.

Response 7: We gave up the figure 6 of the manuscript. Now figure 7 becomes 6 and contains all the necessary information.

Point 8. In Figure 8, the resolution of FT-IR spectra is too low since the baseline of the spectra has some problems.

Response 8: We corrected the FT-IR spectra taking into account the reviewer's recommendations.

We would like to thank for your review for raising the important points and providing with the corresponding suggestions to improve upon the quality of the manuscript.

Yours Sincerely,

Prof. Gabriel Lucian Radu

Reviewer 2 Report

The work presents a very interesting study on the modification on membranes for improving their performance. The work is well performed and the results are sound. The work is publishable after addressing very minor points:

-the correlation between hydrophobicity and fouling in the introduction is not clear, authors should extend its discussion

-the determination of the pore size from the adsorption isotherm should be better explained

-SEM images of membranes decorated with silica nanoparticles suggest that the increase of the temperature enhances the silica incorporation. However, authors claim for a reduction of the particle size. This is not clear at all.

-the determination of the contact angle for evaluating the hydrophilicity of membranes including silica particles is mislesding. The roughness induced by the particles can induce pinning-depinning phenomena of the contact line which deserve much attention

Author Response

Response to Reviewer 2 Comments

The work presents a very interesting study on the modification on membranes for improving their performance. The work is well performed and the results are sound. The work is publishable after addressing very minor points:

Point 1. The correlation between hydrophobicity and fouling in the introduction is not clear, authors should extend its discussion.

Response 1: Fouling is due to the solute-membrane material interactions, causing partial or even total blockage of the pores, resulting in a continuous decline of flow.

Previous studies indicated that the physical-chemical properties of membrane surface, such as hydrophilicity and roughness, are major factors influencing membrane fouling. It is generally accepted that hydrophilic membrane corresponds to lower membrane fouling potential than hydrophobic one because many foulants are hydrophobic in nature (Weis et al., 2005; Rana and Matsuura, 2010). Also, the most hydrophobic NF-membrane, provided the lowest permeate flux in the nanofiltration extract (Cissé et al., 2011).

We have introduced these additions in the lines 37-42.

Point 2. The determination of the pore size from the adsorption isotherm should be better explained.

Response 2: The pore size was not determined from the adsorption isotherms but their shape gives information on the pore size. Thus, the value of relative pressure at the inflection point gives information on the pore size. This is why a sharp inflection between 0.6–0.8 range indicates the presence of mesopores as was specified in the manuscript.

“The textural properties were characterized by N2 sorption experiments. As, we can see that adsorption branch of isotherms (Figure 1b) show a sharp inflection between 0.6–0.8 range of the relative pressure which is typical of mesoporous silica materials like KIT-6 mesoporous silica (Guillet-Nicolas et al., 2021). It can also be seen the preservation of KIT-6 texture with H1 hysteresis loop after functionalization of silica with amine group. The samples shown a narrow pore size distribution centered at 7.2 nm, respectively 6.4 nm. The best Brunauer–Emmett–Teller (BET) specific surface (780 m2/g) was obtained for sample obtained by hydrothermal treated at 100 °C.”

We have introduced the additional information in the lines 144-150.

Point 3. SEM images of membranes decorated with silica nanoparticles suggest that the increase of the temperature enhances the silica incorporation. However, authors claim for a reduction of the particle size. This is not clear at all.

Response 3: Thanks for your suggestion. Thus, we selected new SEM images for the obtained silica powders.  These images show, for the selected KIT-6 powder, a spherical morphology of particles with smaller and more uniform sizes.

A new text has been introduced in this regard in the revised manuscript in the lines 164-168.

Point 4. The determination of the contact angle for evaluating the hydrophilicity of membranes including silica particles is mislesding.

Response 4: Our result is in agreement with other similar studies which showed that the addition of SiO2 nanoparticles in the polymeric matrix tended to decrease membrane contact angles (Sultan et al., 2020). 

We added the commentary in the lines 186-187.

We would like to thank for your review for raising the important points and providing with the corresponding suggestions to improve upon the quality of the manuscript.

Yours Sincerely,

Prof. Gabriel Lucian Radu

Reviewer 3 Report

The manuscript entitled “Nanofiltration composite membranes based on KIT-6 and functionalized KIT-6 nanoparticles in a polymeric matrix with enhanced performances” the authors studied the performances of KIT-6 and functionalized KIT-6 nanoparticles nanofiltration composite membranes in a polymeric matrix. The as-synthesized of KIT-6 and functionalized KIT-6 nanoparticles was confirmed by using various spectroscopic and microscopic techniques such as UV-Vis, FT-IR, AFM and SEM. The manuscript is well organized, and the study is sufficiently performed. The result analysis is very accurate and adequate, lacks of major errors, reference list is appropriate and up-to-dated. Therefore, I would recommend the publication of the manuscript in the “MEMBRANES” after some MINOR improvements.

The comments are provided as follows:

Comment 1. Firstly, I would like to draw attention of the authors that I found there are some typographical errors in the manuscript, so authors need to correct it in the revised manuscript.

Comment 2. Figure 1 resolution is very poor, replace with high resolution figure.

Comment 3. Add the SEM analysis sample preparation details in the characterization Section.

Comment 4. Add the major peak values in the FT-IR spectrum figure for more clear understanding of readers.

Comment 5.  In SEM and AFM results: The authors should explore and discuss better their results with some more references to prepare a better discussion.

Comment 6. The similarity report of the article is *22%*. However, the authors need to rewrite the section 2.3 and 2.4. Please refer to the attached similarity report.

Author Response

Response to Reviewer 3 Comments

Point 1. Firstly, I would like to draw attention of the authors that I found there are some typographical errors in the manuscript, so authors need to correct it in the revised manuscript.

Response 1: We revised the manuscript. The typographical errors were corrected in the revised manuscript.

Point 2. Figure 1 resolution is very poor, replace with high resolution figure.

Response 2: Thanks for the comment. We corrected the Figure 1 (XRD pattern image) for a better resolution.

Point 3. Add the SEM analysis sample preparation details in the characterization Section.

Response 3: We added the new information about SEM analysis in line 108-113.

Point 4. Add the major peak values in the FT-IR spectrum figure for more clear understanding of readers.

Response 4: We corrected the FT-IR spectra taking into account the reviewer's recommendations.

Point 5.  In SEM and AFM results: The authors should explore and discuss better their results with some more references to prepare a better discussion.

Response 5: Thanks for the comment. According to the recommendation, the discussions of SEM images were improved to be in agreement with the selection of the sample with the most appropriate morphology for incorporation into the polymer matrix of the membrane. We added some discussion about SEM result and comparison with other research.

“Thus, SEM images (Fig. 2) show for samples obtained by hydrothermal treatment at 80 °C of larger and more compacted particles. For higher temperature (120 °C) a larger particle size distribution can be observed. Therefore, the optimum temperature of hydrothermal treatment is 100 °C, conditions in which a spherical morphology of particles with smaller and more uniform sizes (Fig. 3a) was obtained for KIT-6 powder.”

The discussion of the AFM results was also completed.

“The results of the surface topography of the membranes, obtained by AFM, are presented in Figures 5. All images represent 5µm x 5µm surface. The bar of the right side of the each image indicates the vertical deviation in the sample. Thus, the highest surface is white and the lowest is darkest. As well, each AFM image of the obtained membranes is accompanied by the profile of the morphological properties of surface. This is represented by distance variation in the surface of the membrane with a line traversing the image.  Therefore, AFM results show, in condition of low silica concentration, insignificant variation of roughness and surface porosity compared with PPEES membrane (M0). The lowest roughness surface was observed for nanofiltration composite membranes with amine functionalized KIT-6 nanoparticles (M1and M3 versus M2 and M4). This may be attributed to better interaction between silica nanoparticles, with more hydrophobic surface, and PPEES polymer.”

Point 6. The similarity report of the article is *22%*. However, the authors need to rewrite the section 2.3 and 2.4. Please refer to the attached similarity report.

Response 6: We rewrite and made additions in the section 2.3 and 2.4.

Round 2

Reviewer 1 Report

I agree with the publication of this manuscript.